# UWB-Assisted Bluetooth Localization Using Regression Models and Multi-Scan Processing

**DOI:** 10.3390/s24196492

**Published:** 2024-10-09

**Authors:** Pan Li, Runyu Guan, Bing Chen, Shaojian Xu, Danli Xiao, Luping Xu, Bo Yan

**Affiliations:** 1School of Aerospace Science and Technology, Xidian University, Xi’an 710126, China; pli@xidian.edu.cn (P.L.); 22131214141@stu.xidian.edu.cn (R.G.); 24131213475@stu.xidian.edu.cn (D.X.); mail2111@163.com (L.X.); 2Laboratory of Transport Safety and Emergency Technology, Transport Planning and Research Institute, Beijing 100029, China; chenbing@tpri.org.cn; 3Beijing Leiyin Electronic Technology Development Co., Ltd., Beijing 100070, China; xushaojian@139.com

**Keywords:** indoor tracking, cross-floor localization, multi-frame fusion particle filtering, UWB

## Abstract

Bluetooth devices have been widely used for pedestrian positioning and navigation in complex indoor scenes. Bluetooth beacons are scattered throughout the entire indoor walkable area containing stairwells, and pedestrian positioning can be obtained by the received Bluetooth packets. However, the positioning performance is sharply deteriorated by the multipath effects originating from indoor clutter and walls. In this work, an ultra-wideband (UWB)-assisted Bluetooth acquisition of signal strength value method is proposed for the construction of a Bluetooth fingerprint library, and a multi-frame fusion particle filtering approach is proposed for indoor pedestrian localization for online matching. First, a polynomial regression model is developed to fit the relationship between signal strength and location. Then, particle filtering is utilized to continuously update the hypothetical location and combine the data from multiple frames before and after to attenuate the interference generated by the multipath. Finally, the position corresponding to the maximum likelihood probability of the multi-frame signal is used to obtain a more accurate position estimation with an average error as low as 70 cm.

## 1. Introduction

In contemporary society, indoor positioning technology has significantly enhanced the functionality and interactivity of various spaces such as indoor navigation [1,2]. The quintessence of this technology lies in its ability to provide precise location data within indoor environments, where traditional Global Positioning Systems (GPSs) falter due to signal attenuation and obstruction [3,4]. The devices for the indoor positioning include Wireless Fidelity (Wi-Fi), Radio Frequency Identification (RFID), Ultrasonic positioning systems, Inertial Navigation Systems (INS), UWB technology, and Bluetooth [5]. Figure 1 organizes the current state of research in indoor positioning in recent years.

Wi-Fi, often available without extra hardware, is cost-effective for indoor positioning [6,7]. It offers moderate accuracy without needing line of sight. However, its accuracy can suffer from signal interference by electronics and physical obstacles. Signal strength varies with device density and network traffic, causing inconsistent results. Wi-Fi positioning’s effectiveness also relies on the existing infrastructure, which may vary in robustness.RFID [8,9] technology is cost-effective and easy to deploy, offering high accuracy and scalability. Its limitations include a short range for passive tags, the need for strategic reader placement, susceptibility to metal and electromagnetic interference, limited data capacity, and security and privacy concerns.Ultrasonic indoor positioning systems are accurate, immune to electromagnetic interference, cost-effective, and safe. However, they have a limited range, are sensitive to environmental factors, and can be affected by obstructions and reflections. Installation and calibration can be complex, and environmental conditions may impact their precision due to variations in sound speed.INS are advantageous for indoor positioning as they are independent of external signals and provide continuous, initially accurate tracking. However, INS cannot determine absolute positions and are prone to accumulating errors over time, and the accuracy and reliability are reduced.UWB technology [10] excels in indoor positioning with its high accuracy, obstacle penetration, and secure, low-interference data transmission. However, its higher cost, limited range, specialized infrastructure needs, and regulatory constraints are significant considerations.Bluetooth is a favorable choice for indoor positioning due to its ubiquity, low energy use, and cost-effectiveness [11]. It provides decent accuracy, is scalable and easy to install. Therefore, it is suitable for various applications like navigation and asset tracking. However, it faces issues with signal interference, physical barriers, and range limitations.

Considering the necessity of a cost-effective, wide-area indoor positioning system for pedestrian navigation, the choice of a Bluetooth system is strategic and well-founded. It offers the required positional accuracy for indoor pedestrian navigation at a fraction of the cost and complexity compared to other systems. The methods involved in Bluetooth-based indoor positioning include:Received Signal Strength Indicator (RSSI) [12,13,14,15,16]: This method calculates the distance between the device and the Bluetooth beacon based on the strength of the received signal. Closer proximity to the beacon results in a stronger signal. The system can accurately determine the device’s position by calculating the intersection of these distances.Angle of Arrival (AoA): These techniques measure the angle at which the signal arrives at or departs from the device or beacon. These data, combined with distance measurements, can enhance the accuracy of the positioning.Fingerprinting [17,18,19,20]: This method involves creating a map of signal strength throughout the building. The system then matches the observed signal strengths from various beacons to this map to locate the device [21].Data Fusion with Other Technologies [22,23,24,25]: Combining Bluetooth data with information from other technologies like Wi-Fi, GPS (where applicable), or inertial sensors in smartphones can lead to more accurate positioning, especially in complex indoor environments.

These systems are useful in various applications like navigation in malls or airports, asset tracking in warehouses, and in context-aware advertising. The accuracy and reliability of Bluetooth-based indoor positioning can vary depending on the environment [26], number of beacons, and the specific technology used.

However, there are still great challenges to achieve high precision position information and trajectories for indoor positioning of pedestrians, and there are six issues that need to be addressed to obtain satisfactory trajectories, which can be summarized as follows:Bluetooth multipath effect. Due to the occlusion of indoor objects and the varied terrain [27], the signal strength will be subject to different situations such as reflection, superposition, and missing, etc.Complex indoor environment. The localization area consists of several relatively independent regions, and each region has its own characteristics, in addition to the need to consider the possibility of object movement.Processing of Bluetooth data. The key features of the data are extracted, the clutter is filtered out, and the missing items are filled in.Layout selection of Bluetooth beacons [28]. Fewer beacons may prevent the device from receiving the Bluetooth signal, thus preventing it from localization, and more new tables will cause interference with each other and reduce accuracy.A large amount of workload for fingerprint library collection. The movement of indoor objects and the requirements of high-precision positioning require time-consuming and irregular updating of the fingerprint library [29].Higher accuracy positioning algorithm. The current K-Nearest Neighbor (KNN) algorithm has an accuracy of about 1.5 m [30], which is not able to meet the current localization requirements and requires a more accurate algorithm for localization.

This study aims to achieve low-cost, high-precision, cross-floor indoor positioning using Bluetooth technology. UWB is utilized to assist Bluetooth in the fingerprint database collection. The collected RSSI data are then analyzed and optimized using Kalman filtering. Finally, the positioning and optimization of the results are carried out by combining regression models, particle filtering, and multi-frame algorithms. The proposed method offers the following advantages:The use of UWB-assisted Bluetooth enables rapid database construction, reducing time and labor costs.Applying Kalman filtering to multi-frame RSSI data reduces multipath interference and improves data accuracy.The combination of regression models, particle filtering, and multi-frame algorithms extracts feature parameters to enhance positioning accuracy and efficiency, and constrain positioning results.

The structure of this paper is as follows: Section 2 provides an overview of the system, including the system model and measurement model. Section 3 describes the specific steps of the proposed method. Section 4 validates the superiority of the method through experiments in complex indoor scenarios. Section 5 summarizes the limitations of the proposed method and outlines future work.

## 2. System Overview and Notation

### 2.1. System Model

The study utilizes a network composed of low-power Bluetooth beacons, which are deployed to determine and localize pedestrian positional data and motion trajectories within designated areas. In this context, AP denotes the Bluetooth beacons present in the area, *N* represents the index of each Bluetooth beacon, *t* signifies time, and *i* indicates the current position of the target.

#### 2.1.1. Target Model

Assuming pedestrians move within the designated area, their state variables at time *t* are represented as $\left(x_{t},y_{t},h_{t},v_{t},a_{t}\right)$, where $\left(x_{t},y_{t},h_{t}\right)$ denotes the current position, $v_{t}$ signifies the current walking speed, and $a_{t}$ indicates the direction or orientation of the pedestrian. The motion model of pedestrians at time *t* can be expressed as follows:(1)$$\begin{array}{l} x_{t+1}=x_{t}+v_{t}\cos(\alpha_{t})\Delta t+\omega_{xt}\\ y_{t+1}=y_{t}+v_{t}\sin(\alpha_{t})\Delta t+\omega_{yt}\\ h_{t+1}=h_{t}+\omega_{ht}\\ v_{t+1}=v_{t}+\omega_{vt}\\ \alpha_{t+1}=\alpha_{t}+\omega_{t}\Delta t+\omega_{\alpha t} \end{array}$$
where Δ*t* represents the time interval, $\omega_{t}$ denotes angular velocity, and $\omega_{xt}$, $\omega_{yt}$, $\omega_{vt}$, $\omega_{at}$ signifies process noise, indicative of system uncertainty.

For example, in a region populated with *N* Bluetooth beacons, each situated at known co-ordinates $\left(x_{i},y_{i},h_{i}\right)$, the distances $d_{i}$ from a pedestrian to each beacon at time *t* can be determined by measuring the corresponding RSSI values. Based on the relationship between RSSI and distance, the following equation is derived:(2)$$d_{i}=\sqrt{{\left(x_{t}-x_{i}\right)}^{2}+{\left(y_{t}-y_{i}\right)}^{2}+{\left(h_{t}-h_{i}\right)}^{2}}$$
this computation provides the pedestrian’s position at each discrete time step.

#### 2.1.2. Measurement Model

A network of low-power Bluetooth beacons is strategically deployed throughout the designated area, each beacon emitting signals in the iBeacon protocol at intervals of 100 ms with a transmission power of 0 dBm. Upon entering this area, a pedestrian’s device, such as a smartphone, captures signals from multiple beacons with varying intensities. The strength of the received signal is inversely proportional to the distance from the beacon: a stronger signal corresponds to a shorter distance, while a complete absence of signal indicates the device’s location is beyond the effective range of the beacon.

The measurement model, depicted in Figure 2, leverages the distance attenuation properties inherent in RSSI. Within a consistent scenario, varying locations exhibit different distances from the AP, leading to unique RSSI values at each position. These variations facilitate positional differentiation and the construction of an offline reference database.

In the operational positioning phase, signal strength data are acquired and compared against the closest entry in the fingerprint database. The algorithm for distance estimation is outlined as follows:(3)$$RSSI=-10n\log_{10}(d)+A$$
(4)$$d_{i}=10^{\frac{A-RSSI_{N}}{10n}}$$
where *RSSI* denotes the received signal strength, *n* denotes the environmental attenuation factor, *A* indicates the signal strength at a distance of 1 m between the transmitter and receiver, *d* signifies the distance between the transmitter and receiver, and *RSSI_N_* specifies the received signal strength from the *N*th access point AP.

### 2.2. Framework

Utilizing UWB Bluetooth technology, an offline fingerprint database is established, which is subsequently integrated with a regression model and multi-Frame fusion particle filtering for real-time pedestrian localization, as illustrated in Figure 3.

A substantial volume of robust Bluetooth signal data is essential for constructing the regression model. Concurrently, environmental factors and inter-beacon interference are addressed to extract pertinent features and iteratively refine model parameters.Jointly analyzing Bluetooth signal data across multiple time points mitigates errors associated with individual time points, thus enhancing localization accuracy. Although this approach demands increased data storage and computational resources, it significantly improves the reliability of the localization trajectory.This phase encompasses offline data collection using UWB-assisted Bluetooth beacons, which reduces both initial time and labor costs. However, due to the inherent variability in indoor environments, it is crucial to regularly update UWB map data to promptly detect and address environmental changes, thereby ensuring the provision of reliable localization map data.

## 3. Processing

This study consists of three main components: the first component focuses on utilizing UWB technology instead of manually assisted Bluetooth for data acquisition, the second component involves the determination of a regression model, and the last part is localisation by multi-frame fusion particle filtering.

### 3.1. UWB-Assisted Bluetooth Map Building

In the offline stage of Bluetooth fingerprint library localization, it is common to arrange multiple Bluetooth beacons in the area of interest and divide the area into grids. Points are then uniformly selected within the area, and their position co-ordinates and RSSI sequences are recorded to establish a comprehensive fingerprint library. To enhance the efficiency and accuracy of fingerprint data collection, this study incorporates UWB technology to assist in Bluetooth map building. A remote-controlled cart equipped with UWB and Bluetooth collection modules is used for automated data collection. This approach improves the efficiency of data collection without compromising positioning accuracy, the specific process is shown in Figure 4.

#### 3.1.1. UWB Node Deployment

The UWB nodes are typically installed at fixed locations within a building to cover the entire area of interest for localization. The number and placement of these nodes are determined based on the building’s structure and positioning requirements, ensuring comprehensive coverage of the indoor area. Figure 5 illustrates the deployment, where the red rectangular boxes represent the UWB base stations. These base stations are arranged at the four corners of the area, allowing the targets within the area to receive signals transmitted by the base stations. The received signals are then processed and provided as feedback to the base stations themselves.

#### 3.1.2. Distance Measurement

In this paper, the UWB localization experiment utilizes the trilateral ranging localization method. As depicted in Figure 6, the numbers 1, 2, and 3 represent three distinct UWB base stations involved in localization, while *X* represents the UWB tag. By employing the DS-TWR ranging method, the distances between the three base stations and the tag *X* can be measured. Subsequently, the distance equations of these three base stations can be solved to determine the location co-ordinates of the tag. The principle behind this method is illustrated in Figure 7. Geometrically, the three circles formed based on the ranging information and the ranging centers will intersect at a single point, which corresponds to the calculated location of the tag.

Assuming that the co-ordinates of base stations 1, 2, 3 and the tag are $\left(x_{1},y_{1}\right)$, $\left(x_{2},y_{2}\right)$, $\left(x_{3},y_{3}\right)$ and (x,y) respectively, the following equation can be derived:(5)$$\left\{ \begin{array}{l} {\left(x-x_{1}\right)}^{2}+{\left(y-y_{1}\right)}^{2}={r_{1}}^{2}\\ {\left(x-x_{2}\right)}^{2}+{\left(y-y_{2}\right)}^{2}={r_{2}}^{2}\\ {\left(x-x_{3}\right)}^{2}+{\left(y-y_{3}\right)}^{2}={r_{3}}^{2} \end{array}\right.,\;AX=B$$
among them,
(6)$$X=\left[\begin{array}{l} x\\ y \end{array}\right],\;A=\left[\begin{array}{cc} 2(x_{1}-x_{3}) & 2(y_{1}-y_{3})\\ 2(x_{2}-x_{3}) & 2(y_{2}-y_{3}) \end{array}\right],\;B=\left[\begin{array}{l} {x_{1}}^{2}-{x_{3}}^{2}+{y_{1}}^{2}-{y_{3}}^{2}-{r_{1}}^{2}\\ {x_{2}}^{2}-{x_{3}}^{2}+{y_{2}}^{2}-{y_{3}}^{2}-{r_{2}}^{2} \end{array}\right]$$

The two-dimensional co-ordinates of the tag can be obtained by solving the system of equations.

#### 3.1.3. Map Construction

Based on the distance information obtained from the UWB nodes, the localization system initiates the construction of an indoor environment map [31]. The setup for data collection is illustrated in Figure 8, where the central area is occupied by the mobile collection equipment represented by a red rectangular box. Surrounding the equipment are four UWB base stations labeled as A, B, C, and D. During the data collection process, the mobile device remains stationary at various locations within the designated area. At each location, it captures the Bluetooth RSSI signal data, which is subsequently filtered and processed. This data processing step generates fingerprint points representing the current position. The collection of all fingerprint points obtained at different positions forms an offline fingerprint library.

#### 3.1.4. UWB and Bluetooth Signal Fusion

By integrating the map information obtained from the UWB technology, the localization system enhances its ability to correct errors in Bluetooth signals. This fusion of data allows for a more precise and reliable localization of the device’s position. The UWB technology provides additional information that helps improve the accuracy and reliability of the localization system by compensating for any potential inaccuracies or inconsistencies in the Bluetooth signals.

### 3.2. Positioning Algorithm Based on Regression Model

To begin, the determination of the regression model is a crucial step in our algorithm, as shown in Figure 9. In this paper, we employ polynomial regression to analyze the non-linear relationship between variables. The polynomial interpolation theorem supports our approach, indicating that for any continuous function, a polynomial function can be found under specific conditions. This polynomial function closely aligns with the original function within a certain range. By utilizing polynomial regression, we can effectively capture and model the complex relationships between variables, improving the accuracy of our positioning algorithm.

#### 3.2.1. Least Squares Fitting Bluetooth Fingerprint Library

Input the fingerprint library and perform least squares to extract the features of the fingerprint library and determine the various parameters of the polynomial: in order to extract the features of the fingerprint library and determine the various parameters of the polynomial, we utilize the method of least squares. Least squares is a widely used mathematical fitting technique that aims to find the best fit for the model parameters by minimizing the error between observed and predicted values. It achieves this by seeking the derivative of the error function to approximate the minimum value. This method is known for its simplicity in thinking, mathematical comprehensibility, and effectiveness in estimating model parameters. Least squares is commonly employed in solving regression model parameters and is particularly suitable for our analysis of the fingerprint library.

Assuming that J(w) is minimal when $W=\hat{W}$, we can obtain the following by finding the maximum value of J(w) differentiation:(7)$$X^{T}X\overset{\wedge }{W}=X^{T}Y$$
if the matrix is non-singular, there is a unique solution:(8)$$\overset{\wedge }{W}={\left(X^{T}X\right)}^{-1}X^{T}Y$$

#### 3.2.2. Determine Regression Parameters and Model

An initial model is determined and the parameters obtained in the previous step were used to iterate the model eigenvalues to obtain the regression model and regression parameters: To obtain the regression model and regression parameters, an initial model is determined using the parameters obtained in the previous step. Polynomial regression is chosen as it is highly applicable in this context. The general binary quadratic polynomial equation is given by:(9)$$y_{i}=w_{0}^{i}+w_{1}^{i}x_{1}^{i}+w_{2}^{i}x_{2}^{i}+w_{3}^{i}x_{1}^{i}x_{2}^{i}+w_{4}^{i}{\left(x_{1}^{i}\right)}^{2}+w_{5}^{i}{\left(x_{2}^{i}\right)}^{2}$$
where $\left(x_{1}^{i},x_{2}^{i}\right)$ represents the independent variable and $y_{i}$ represents the dependent variable, and w is the parameter of the polynomial.

RSSI decays continuously as the distance between the receiving end and the transmitting end increases. This decay relationship is non-linear and cannot be adequately fitted by linear regression models or classification models. Therefore, a polynomial equation is chosen to model the relationship between the co-ordinates of the receiving end position and the corresponding RSSI. Assuming there are *k* Bluetooth beacons and RSSI data for *n* positions, with the co-ordinates $\left(x_{1},y_{1}\right)$ as the independent variable and the RSSI as the dependent variable, the above equation can be expressed in matrix form as follows:
(10)$$\begin{array}{l} Y=XW,J(w)=\sum_{i=1}^{n}{\left(y_{i}-w^{i}x^{i}\right)}^{2},Y=\left[\begin{array}{c} y_{1}\\ y_{2}\\ \ldots \\ y_{n} \end{array}\right]=\left[\begin{array}{cccc} {{RSSI}_{1}}^{1} & {{RSSI}_{2}}^{1} & \ldots & {{RSSI}_{k}}^{1}\\ {{RSSI}_{1}}^{2} & {{RSSI}_{2}}^{2} & \ldots & {{RSSI}_{k}}^{2}\\ \ldots & \ldots & \ldots & \ldots \\ {{RSSI}_{1}}^{n} & {{RSSI}_{2}}^{n} & \ldots & {{RSSI}_{k}}^{n} \end{array}\right]\\ X=\left[\begin{array}{cccccc} 1 & {x_{1}}^{1} & {x_{2}}^{1} & {x_{1}}^{1}{x_{2}}^{1} & {\left({x_{1}}^{1}\right)}^{2} & {\left({x_{2}}^{1}\right)}^{2}\\ 1 & {x_{1}}^{2} & {x_{2}}^{2} & {x_{1}}^{2}{x_{2}}^{2} & {\left({x_{1}}^{2}\right)}^{2} & {\left({x_{2}}^{2}\right)}^{2}\\ \ldots & \ldots & \ldots & \ldots & \ldots & \ldots \\ 1 & {x_{1}}^{n} & {x_{2}}^{n} & {x_{1}}^{n}{x_{2}}^{n} & {\left({x_{1}}^{n}\right)}^{2} & {\left({x_{2}}^{n}\right)}^{2} \end{array}\right],W=\left[\begin{array}{cccc} {w_{0}}^{1} & {w_{0}}^{2} & \ldots & {w_{0}}^{k}\\ {w_{1}}^{1} & {w_{1}}^{2} & \ldots & {w_{1}}^{k}\\ {w_{2}}^{1} & {w_{2}}^{2} & \ldots & {w_{2}}^{k}\\ {w_{3}}^{1} & {w_{3}}^{2} & \ldots & {w_{3}}^{k}\\ {w_{4}}^{1} & {w_{4}}^{2} & \ldots & {w_{4}}^{k}\\ {w_{5}}^{1} & {w_{5}}^{2} & \ldots & {w_{5}}^{k} \end{array}\right] \end{array}$$
here, the loss function J(w) represents the error between the predicted value and the actual value. To achieve the best fit, the parameter matrix W needs to be solved in order to minimize this error. The regression model is updated by inputting multiple fingerprint libraries and repeating the above steps until the fingerprint library features and the regression model are determined.

Figure 10 depicts the fitting effect of the three Bluetooth AP fingerprint databases. It covers the majority of the fingerprint database data, with an average fitting error of 6.81 dBm. This provides a solid foundation for subsequent particle filtering and multi-frame joint matching.

### 3.3. Localization Using Multi-Frame Fusion Particle Filter

Particle filtering is a state estimation algorithm that relies on random sampling. It employs a set of particles generated randomly to approximate the distribution of the system’s state. These particles are drawn uniformly or according to prior distributions in the state space. The algorithm then updates and estimates the system’s state based on the state equations and measurement equations. Multi-frame fusion, on the other hand, involves analyzing features using data from multiple time points preceding and succeeding the current time. This analysis helps constrain the output results at the current time and minimize error value.

#### 3.3.1. Particle Filtering Processing for Bluetooth Fingerprints

To process the input RSSI sequence, we establish the position hypothesis as the initial particle, with the initial positions randomly distributed in the pre-established fingerprint library. Next, we utilize a constructed polynomial model to calculate the Bluetooth signal strength of each particle. We assign weights to the particles based on the deviation between the calculated value and the actual value. The assigned weights are inversely proportional to the deviation, with higher deviations resulting in smaller weights and lower probability of establishment. Subsequently, we update the set of positional hypotheses, removing hypotheses with lower likelihood probabilities and using the position with a higher likelihood probability as the center to generate a new set of positional hypotheses. Finally, the iteration is terminated when the particle center position no longer changes or the number of iterations reaches a set value. The hypothesis position with the highest likelihood probability is then taken as the calculation result.
Expected signal strength calculation: for each particle position $x_{i}$, we calculate its expected signal strength using the following formula:
(11)$$bl_{est1k}(x_{i})=Amplitude_{cal}(\mathrm{var},x_{i})$$
where $x_{i}$ represents the current position of particle *i*, *var* is a signal-related parameter, and *k* denotes the dimension of the signal. This formula $bl_{est1k}(x_{i})$ indicates the expected signal strength of particle $x_{i}$ in the *k*th dimension.
Distance metric: we calculate the difference between the expected signal strength at the particle location and the actual observed signal strength using the following formula:
(12)$$dis_{part1}(x_{i})=\sum_{k}\left|on_{this1k}-bl_{est1k}(x_{i})\right|$$
where $on_{this1k}$ represents the observed signal strength in frame *i*, $bl_{est1k}$ represents the expected signal strength of particle $x_{i}$ in the *k*th dimension, and $dis_{part1}(x_{i})$ denotes the signal strength difference of $x_{i}$ in frame *i*.
Weight update: we update the weights of the particles based on the measurement model using the following formula:
(13)$$w_{i}\propto \exp\left(-\frac{dis_{part1}(x_{i})}{2\sigma^{2}}\right)$$
where $w_{i}$ is the weight of particle $x_{i}$, and *σ* denotes the standard deviation of the measurement noise.
Particle resampling: based on the updated weights, we select a new set of particles. By sorting the distances, we choose the top 30 particles with the largest weights. Then, we generate new particles around these selected particles.

#### 3.3.2. Optimization of Positioning Results Using Multiple Frames

To improve the accuracy and reliability of positioning, a multi-frame optimization approach is proposed. Firstly, the RSSI sequence of the target position is inputted. After obtaining the initial particle swarm or the particle swarm after resampling, the approximate co-ordinates of the first two positions are estimated for each particle based on the positional assumption. Then, the corresponding RSSI sequences for these three positions are calculated using a polynomial model.

Secondly, the actual input RSSI sequences of the two positions prior to the target point are extracted. By comparing the calculated RSSI sequences from the estimated positions with the actual input RSSI sequences, the signal difference of the multi-frame positions is evaluated to assess the probability of the current position assumption being accurate. This evaluation process is illustrated in Figure 11.

The multi-frame fusion technique utilizes the calculation results of previous positions to constrain the current position, thereby effectively reducing the localization error. The calculation formula is as follows:(14)$$dis(x_{i})=\frac{dis00(x_{i})+dis0(x_{i})+dis1(x_{i})+dis2(x_{i})}{4}$$
where *dis*00(*x_i_*), *dis*0(*x_i_*), *dis*1(*x_i_*), *dis*2(*x_i_*) represent the RSSI data of the top four position hypotheses with the highest probability of being accurate, obtained through resampling of particle $x_{i}$, *dis*(*x_i_*) represents the average RSSI data of particle $x_{i}$ after multi-frame processing.

## 4. Case Study

### 4.1. Experimental Environment

To analyze the improvement brought by the proposed matching algorithm compared to traditional algorithms, experiments were conducted in the experimental building of a university in Xi’an, as depicted in Figure 12 and Figure 13. The experimental site included a first floor area of 24 m × 8 m and a second floor area of 26 m × 8 m. The experiments were conducted in Building G, comprising the first floor, stairwells, and the second floor. A total of 13, 4, and 12 beacons were deployed on the first floor, stairwells, and second floor, respectively, spaced every 4 m with a height of 2 m. The experimental scenario covered approximately 300 m^2^ of irregular, multi-level space with multiple independent areas.

### 4.2. In Comparison with KNN-Based Algorithms

The KNN algorithm can be directly applied to new datasets for classification without the need for pre-training a fingerprint database. It calculates the distance between the sample and the dataset, classifying the K nearest samples into one category. In fingerprint matching localization, the algorithm calculates the difference between the offline database and the actual collected RSSI, selects the K samples with the smallest differences, and determines the coordinates of the target location through the mean of these samples’ position coordinates.

The localization results are depicted in Figure 14 and Figure 15. The analysis of localization results is presented in Table 1. It can be observed that the proposed matching algorithm yields localization results with considerable accuracy, surpassing the KNN/WKNN algorithms [32,33]. The overall average error is 1.13 m, which meets the requirements for daily life localization, demonstrating the feasibility of achieving large-scale continuous indoor Bluetooth localization.

For the trajectory point co-ordinates (5, 2.4), the co-ordinates predicted by the KNN/WKNN algorithms are 5.2 and 4.5, while the co-ordinates predicted by the algorithm in this paper are 4.8 and 2.3. The errors for both are 2.11 m and 0.22 m, respectively. For the trajectory point co-ordinates (15.4, 2.4), the KNN/WKNN algorithms predict the co-ordinates as (16, 4.3), while this paper’s algorithm predicts them as (15.3, 2.5). The errors for these points are 1.99 m and 0.14 m, respectively. Additionally, the y co-ordinate deviation of the KNN/WKNN algorithms is significant, while the algorithm proposed in this paper shows only a slight deviation. Additionally, the execution times for the two methods are 47.32 ms and 25.77 ms, respectively, indicating that the latter has higher operational efficiency.

From the above figure, it can be seen that the trajectory obtained by the KNN/WKNN algorithm [32,33] is more chaotic and cannot reflect the user’s real trajectory. While the shape and contour of the trajectory obtained by the algorithm in this paper are more obvious and very close to the real trajectory.

KNN method is simple and easy to implement, but it is very easily affected by RSSI fluctuation, WKNN algorithm chooses weighting to improve, but the effect is small. In this paper, the algorithm is based on the particle filtering solution, the introduction of multi-frame joint strategy for optimization, joint before and after the calculation of multiple positions, and this strategy on the current positioning results for the constraints prevents the occurrence of the position of the jump, to a certain extent, and attenuates the impact of the fluctuations of the RSSI. Combined with the experimental comparison results in this section, it is proved that the algorithm proposed in this paper improves the performance in all aspects compared with the traditional algorithms KNN and WKNN, and achieves the expected results.

### 4.3. Comparison before and after Kalman Filtering

The Kalman filtering algorithm is a mathematical method used to estimate the states of dynamic systems, widely applied in signal processing and control systems. It provides optimal estimates of the system states by combining prior information about the system with measurement data. The algorithm consists of two main steps: prediction and update. In the prediction phase, the algorithm uses the previous state to estimate the current state; in the update phase, it corrects the prediction using the current measurement values. This method is particularly effective in handling noisy measurement data, allowing for significant reductions in uncertainty and improvements in localization accuracy.

The filtering results for a randomly selected Bluetooth AP are shown in Figure 16, with positioning errors detailed in Table 2. During the mapping phase, the proposed RSSI signal filtering method can reduce mapping signal errors, thereby improving positioning accuracy.

As can be seen from the Figure and Table, the sliding average filter smooths the data only by a simple averaging operation, which reduces random noise but does not distinguish between systematic and measurement noise; the Gaussian filter uses a Gaussian function for weighted averaging but also lacks refinement of the noise; and the Kalman filter, through a Kalman gain computation, allows the filter to make trade-offs between different observations and predictions, thus providing a good estimate in the presence of noise can still provide a better estimate. Therefore, Kalman filtering is the most effective, and the processed data have smaller variance and range of fluctuation compared to the other two filtering methods, which can effectively remove outliers and reduce the interference of noise.

### 4.4. Overall Tracking Performance

The impact of UWB-assisted database construction on Bluetooth positioning accuracy is illustrated in Figure 17 and Figure 18. As shown, UWB positioning demonstrates high accuracy, with an average error within 10 cm. The obtained trajectory closely aligns with the actual trajectory.

Figure 18 infers that the difference in positioning results between UWB-assisted and manually constructed databases is acceptable, with an average error difference of less than 4% and a variance difference of less than 3%. This indicates that the introduction of UWB improves the efficiency of database construction without compromising the positioning accuracy of Bluetooth fingerprints. The impact on accuracy is negligible, demonstrating the feasibility of the UWB-assisted database construction method for automatic database creation.

## 5. Conclusions

Nowadays, satellite signals have serious attenuation indoors and cannot provide effective positioning services. In this paper, Bluetooth fingerprint positioning with low power consumption and considerable accuracy was selected as the research object: for the offline collection phase of Bluetooth fingerprint positioning, it was proposed to introduce UWB as an auxiliary positioning means to complete the collection work, which not only saves a lot of time, but also improves the degree of freedom of the offline collection work, and the error is within 10 cm. The Bluetooth RSSI data were processed by kalman filtering, which controls the fluctuation range within 3 dBm and reduces the noise interference; for the online positioning stage, the matching algorithm was proposed to use the regression model to fit the fingerprint database data and the position assumption by particle filtering, and in order to further improve the positioning accuracy, the calculation of multiple frames before and after the proposed joint calculation was carried out to solve the problem of the positional jumps, through experiments, in single-floor indoor positioning, the positioning accuracy of the first and second floors can reach 0.94 m and 1.70 m respectively, and in the multi-floor indoor positioning, the positioning accuracy can reach 1 m, which reduces error by 23.5% compared to existing methods. However, it is still challenging to achieve seamless switching between floors and higher accuracy of positioning, therefore, in future research work, Bluetooth can be combined with other positioning technologies [34,35] for more accurate positioning.

## Figures and Tables

**Figure 1 sensors-24-06492-f001:**
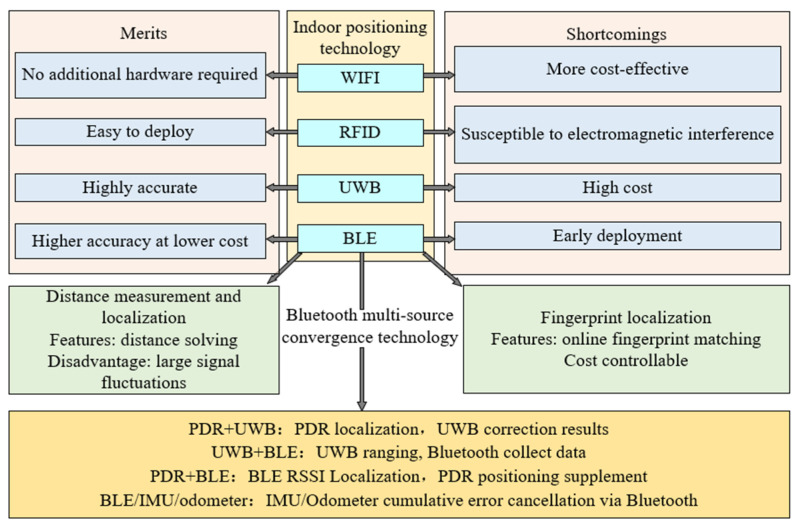
The comparison of other devices and current positioning methods.

**Figure 2 sensors-24-06492-f002:**
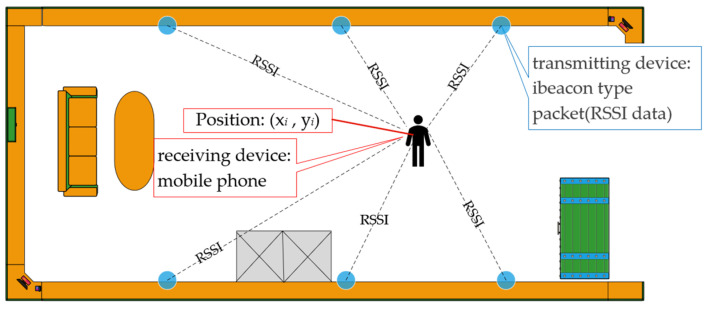
Bluetooth localization measurement model.

**Figure 3 sensors-24-06492-f003:**
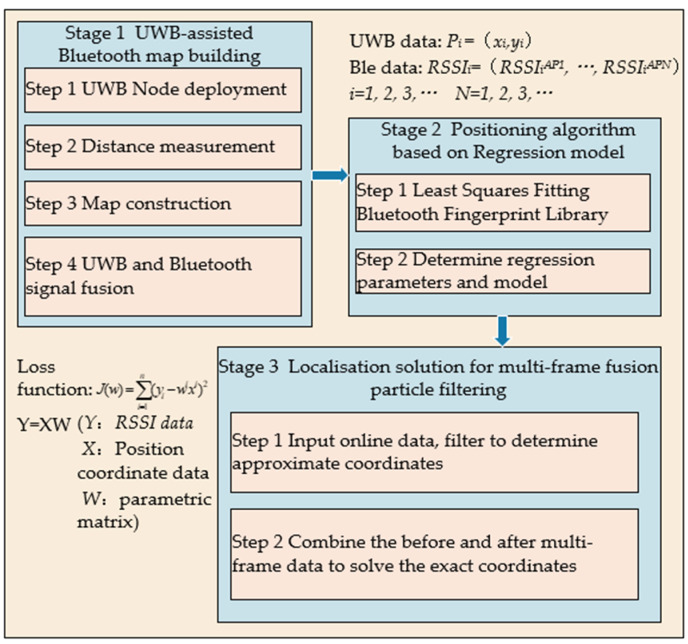
UWB collection and Bluetooth localization flowchart.

**Figure 4 sensors-24-06492-f004:**
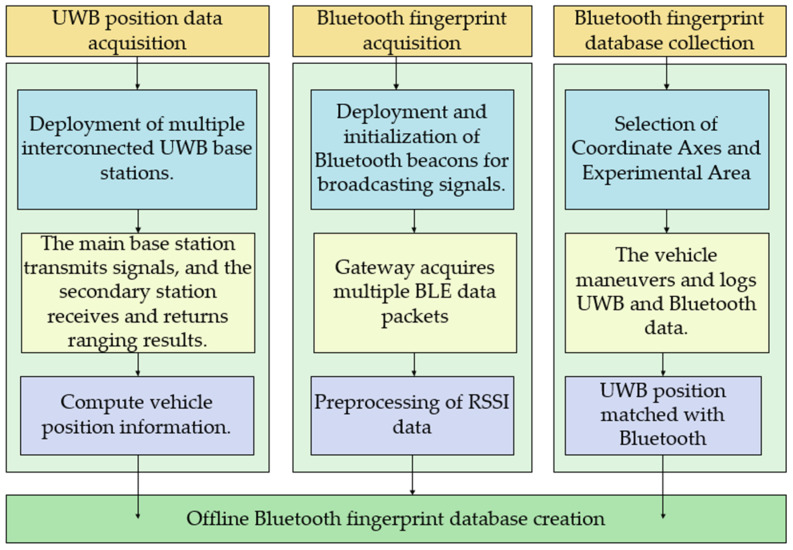
UWB-assisted Bluetooth offline fingerprint database.

**Figure 5 sensors-24-06492-f005:**
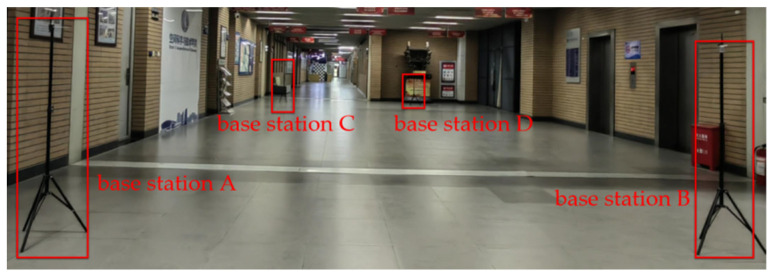
UWB node deployment.

**Figure 6 sensors-24-06492-f006:**
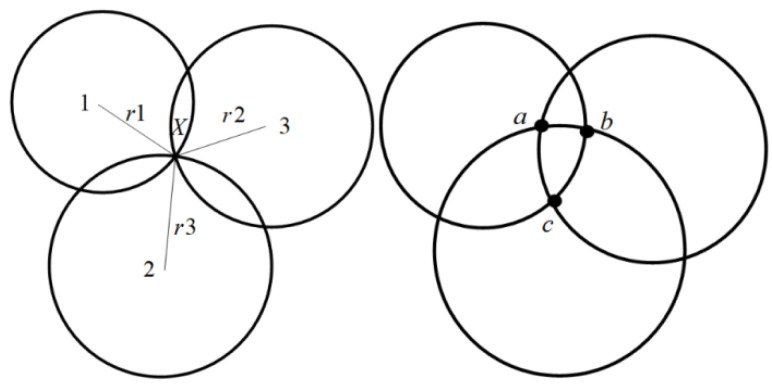
Distance triangulation.

**Figure 7 sensors-24-06492-f007:**
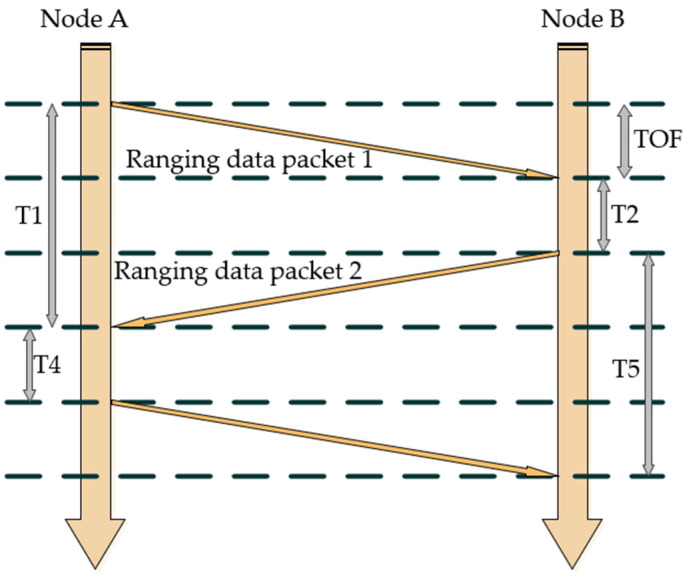
DS-TWR ranging method.

**Figure 8 sensors-24-06492-f008:**
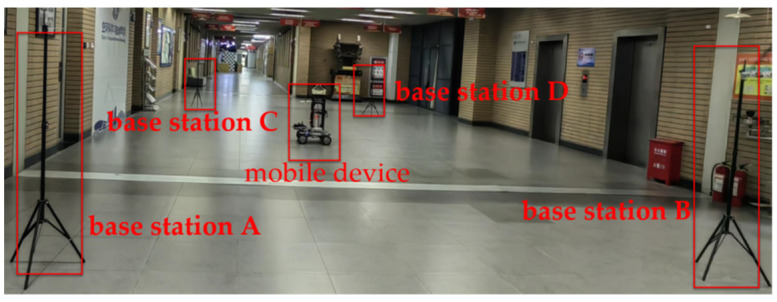
Fingerprinting using mobile devices.

**Figure 9 sensors-24-06492-f009:**
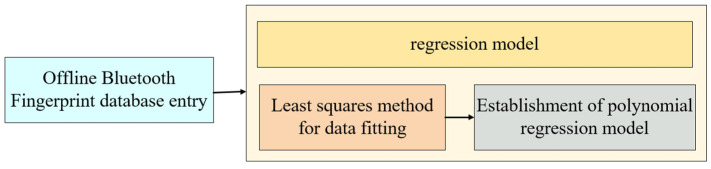
Determination of regression model.

**Figure 10 sensors-24-06492-f010:**
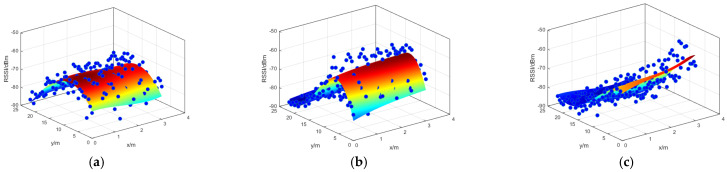
(**a**) AP1. (**b**) AP2. (**c**) AP3.

**Figure 11 sensors-24-06492-f011:**
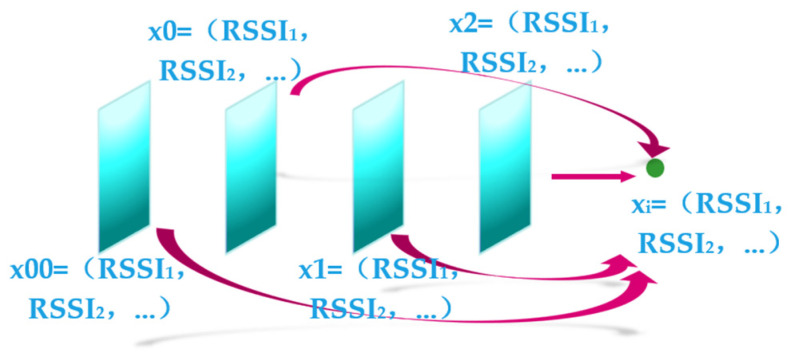
Multi-frame schematic.

**Figure 12 sensors-24-06492-f012:**
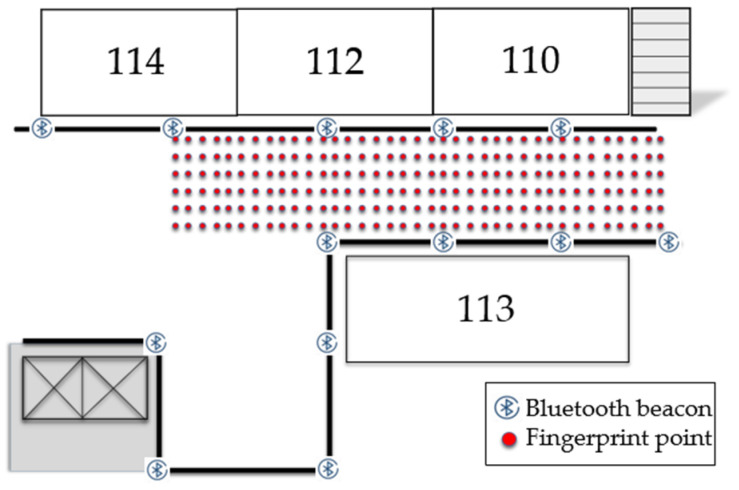
First floor Bluetooth setup.

**Figure 13 sensors-24-06492-f013:**
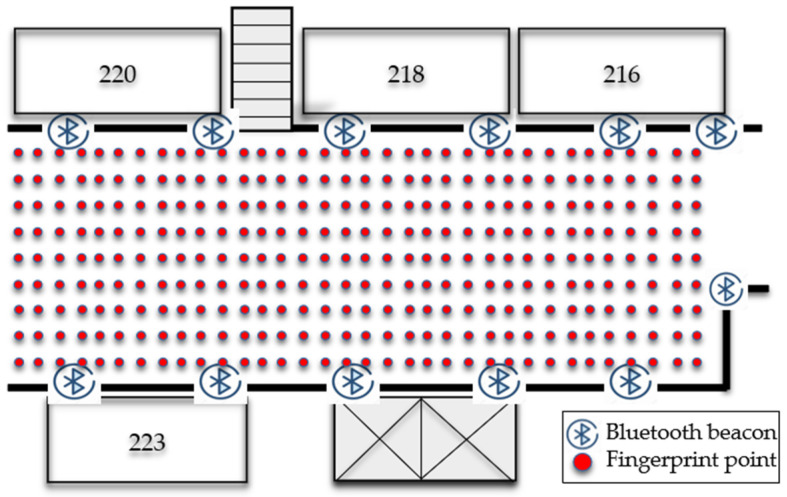
Second floor Bluetooth setup.

**Figure 14 sensors-24-06492-f014:**
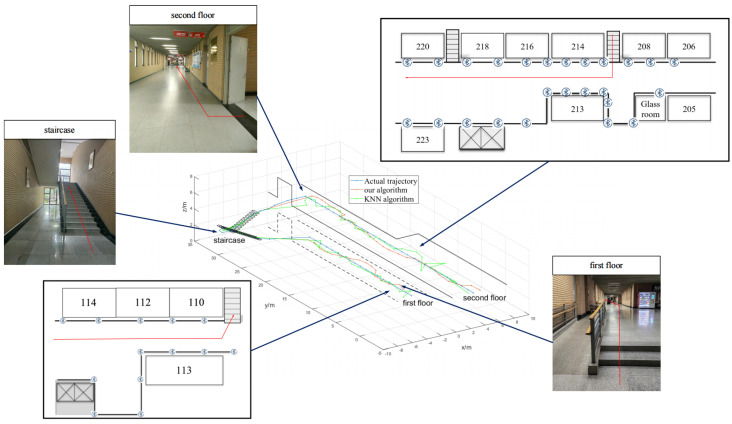
Positioning result.

**Figure 15 sensors-24-06492-f015:**
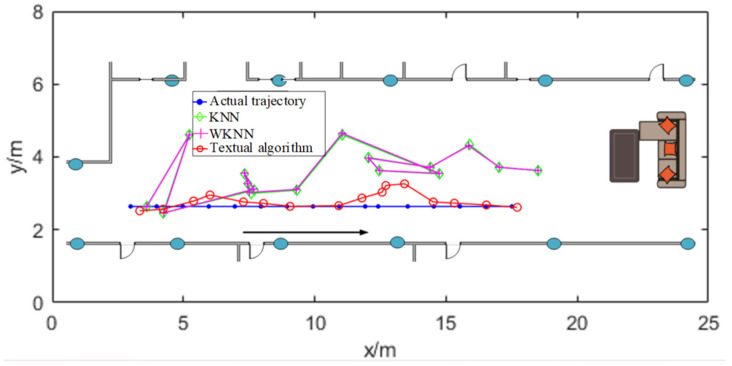
Comparison of localization trajectories.

**Figure 16 sensors-24-06492-f016:**
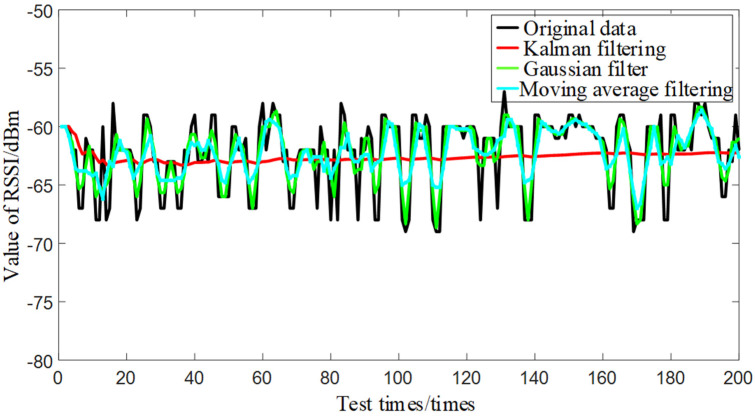
Comparison of filtering methods.

**Figure 17 sensors-24-06492-f017:**
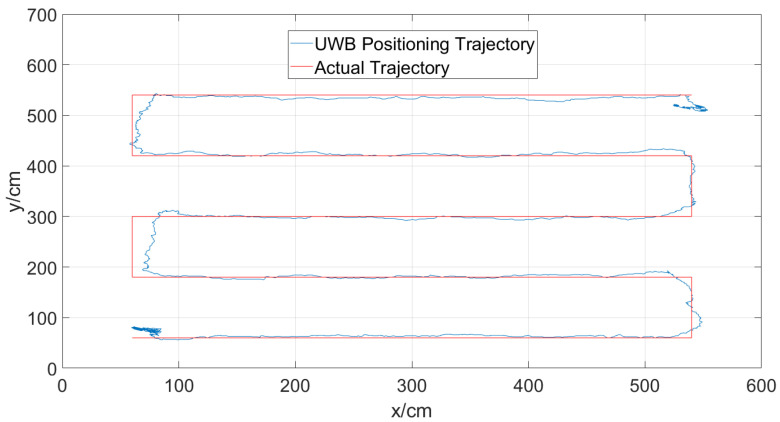
UWB localization results.

**Figure 18 sensors-24-06492-f018:**
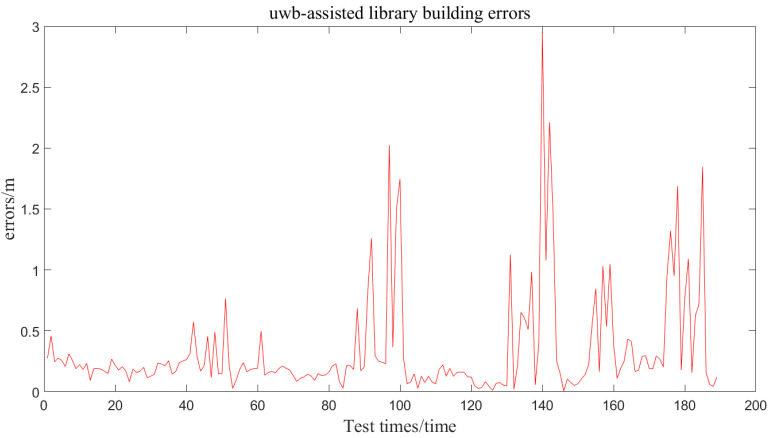
UWB localization error.

**Table 1 sensors-24-06492-t001:** Average localization error.

Positioning Algorithm	First Floor	Second Floor
KNN [32]	0.95 m	1.71 m
WKNN [33]	0.94 m	1.70 m
OURS	0.71 m	1.31 m

**Table 2 sensors-24-06492-t002:** Average filtering error.

Filtering Method	Mean	Variance	Range of Fluctuation
Original Data	−69.48 dBm	7.40 dBm^2^	12.00 dBm
Moving Average	−69.48 dBm	2.54 dBm^2^	8.00 dBm
Gaussian Filter	−69.47 dBm	3.91 dBm^2^	8.67 dBm
Ours	−69.09 dBm	0.24 dBm^2^	2.97 dBm

## Data Availability

The dataset used in this paper cannot be easily obtained because the data requires pre-installation of hardware equipment layout and post-debugging of related information, and the acquisition process is relatively time-consuming. If you need to access the dataset please send a request to 1359578767@qq.com.

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
