# Peer review of "UWB-Assisted Bluetooth Localization Using Regression Models and Multi-Scan Processing"

_sensors, 2024, doi:10.3390/s24196492_

Round 1
Reviewer 1 Report
Comments and Suggestions for Authors
The paper is nice on my view. I didn't detect any major issue, except this one:
Motion of objects is assumed to be two-dimensional: Eq. (1) and (2). Even in buildings, you can have uneven level floors, etc..., adding a third dimension. What happens in such cases? What is the robustness of your method when you have for example a jump of 20 cm in the vertical direction?
Kalman filtering is NOT a method to reduce noise (line 438)! It is an ESTIMATION method that provides, besides estimated values, estimated noise levels on data and parameters. In this way, it is better than KNN and other "Neural" methods, that are not able to produce estimates on the levels of noise on data and parameters, but only output estimated parameters. So, please correct and cite textbooks.
Besides, there is problems with the high presence of jargon. There is a lot of acronyms, so please put a table at the end listing in plain English their signification: UWB, KNN, etc... probably at least 20 of them throughout the paper. Papers are not only for highly specialized folks.
Reviewer 2 Report
Comments and Suggestions for Authors
Detailed comments are listed below:
1. Some existing works may provide more enlightenment for source localization. For example, Dong et al. (Velocity-Free MS/AE Source Location Method for Three-Dimensional Hole-Containing Structures. Engineering, 2020. 6(7): 827-834) proposed the velocity-free MS/AE source location method utilizes the improve A* search method, which can realize the high-accuracy locating requirements in complex three-dimensional hole-containing structures. Al-Jumaili et al. (Acoustic emission source location in complex structures using full automatic delta T mapping technique. Mechanical Systems and Signal Processing, 2016. 72 (2016): 513-524) proposed a fully automatic delta T mapping technique. Using the Bluetooth fingerprint library to find the distance is similar to using Delta T to find sources.
2. Figure 15 has redundant graphs. Suggest deleting graph (a) and (b) since graph (c) has shown the localization trajectories of two methods.
3. KNN/WKNN algorithms is compared with the proposed method in this manuscript. It is recommended to discuss why KNN/WKNN algorithms perform worse than the proposed method. Some points of trajectories in Figure 15(c) may provide examples for the explanation.
4. Number of the formula between Lines 355 and 356 is incorrect. Therefore, the equations after Eq. (12) should be checked and corrected.
5. It may be easier to understand if you don't use underscores in Eqs. (12) , (13) and (16).
6. How does the computational time of this method compare to the KNN/WKNN algorithm?
